# An Integrated Bioinformatics Approach to Identify Network-Derived Hub Genes in Starving Zebrafish

**DOI:** 10.3390/ani12192724

**Published:** 2022-10-10

**Authors:** Amin Mortazavi, Mostafa Ghaderi-Zefrehei, Mustafa Muhaghegh Dolatabady, Mahdi Golshan, Sajad Nazari, Ayeh Sadat Sadr, Saeid Kadkhodaei, Ikhide G. Imumorin, Sunday O. Peters, Jacqueline Smith

**Affiliations:** 1Cold-Water Fishes Genetic and Breeding Research Center, Iranian Fisheries Research Institute, Agriculture Research Education and Extension Organization (AREEO), Yasouj 74394-75918, Iran; 2Department of Animal Science, Faculty of Agriculture, Yasouj University, Yasouj 74394-75918, Iran; 3Iranian Fisheries Research Institute, Agriculture Research Education and Extension Organization (AREEO), Tehran 14968-13151, Iran; 4South of Iran Aquaculture Research Institute, Iranian Fisheries Science Research Institute (IFSRI), Agricultural Research Education and Extension Organization (AREEO), Ahvaz 19395-1113, Iran; 5Agricultural Biotechnology Research Institute of Iran (ABRII), Center of Iran, Isfahan 14968-13151, Iran; 6School of Biological Sciences, Georgia Institute of Technology, Atlanta, GA 30332, USA; 7Department of Animal Science, Berry College, Mount Berry, GA 30149, USA; 8The Roslin Institute and Royal (Dick), School of Veterinary Studies, The University of Edinburgh, Easter Bush, Midlothian EH25 9RG, UK

**Keywords:** zebrafish, starvation, Bayesian network, co-expression, hub genes, bioinformatic analysis

## Abstract

**Simple Summary:**

In this study, in order to provide a better understanding and new knowledge about biochemical phenomena and gene interactions in the zebrafish transcriptome, the notion of integrative bioinformatics was used. To do so, we used co-expression and protein–protein interaction (PPI) networks followed by Bayesian network (BN) construction. We provide an integrative bioinformatics pipeline to explore crucial transcriptional insights from zebrafish starvation-induced transcriptome data in a user-friendly way with minimal programming skill requirements. In this way, many genes, e.g., *skp1*, *atp5h*, *ndufb10*, *rpl5b*, *zgc:193613*, *zgc:123327*, *zgc:123178*, *wu:fc58f10*, *zgc:111986*, *wu:fc37b12*, *taldo1*, *wu:fb62f08*, *zgc:64133* and *acp5a*, were identified as hub genes affecting gene expression in the liver of starving zebrafish. These results can be applied in many different areas of zebrafish genomic studies.

**Abstract:**

The present study was aimed at identifying causative hub genes within modules formed by co-expression and protein–protein interaction (PPI) networks, followed by Bayesian network (BN) construction in the liver transcriptome of starved zebrafish. To this end, the GSE11107 and GSE112272 datasets from the GEO databases were downloaded and meta-analyzed using the MetaDE package, an add-on R package. Differentially expressed genes (DEGs) were identified based upon expression intensity N(µ = 0.2, σ^2^ = 0.4). Reconstruction of BNs was performed by the bnlearn R package on genes within modules using STRINGdb and CEMiTool. *ndufs5* (shared among PPI, BN and COEX), *rps26*, *rpl10*, *sdhc* (shared between PPI and BN), *ndufa6*, *ndufa10*, *ndufb8* (shared between PPI and COEX), *skp1*, *atp5h*, *ndufb10*, *rpl5b*, *zgc:193613*, *zgc:123327*, *zgc:123178*, *wu:fc58f10*, *zgc:111986*, *wu:fc37b12*, *taldo1*, *wu:fb62f08*, *zgc:64133* and *acp5a* (shared between COEX and BN) were identified as causative hub genes affecting gene expression in the liver of starving zebrafish. Future work will shed light on using integrative analyses of miRNA and DNA microarrays simultaneously, and performing in silico and experimental validation of these hub-causative (CST) genes affecting starvation in zebrafish.

## 1. Introduction

Many fish species experience natural periods of starvation in their habitat due to environmental evolution, seasonal food shortage, migration, transportation, food competition, reproduction and climate change [1,2,3,4,5,6]. In aquaculture, short-term starvation is considered as a strategy to control water quality, reduce handling stress, reduce disease-causing fish mortality, decrease feed costs, increase profitability, increase product quality and decrease flesh lipid content [4,7]. The body’s first tangible response to starvation in fish is weight loss, especially in the first week of starvation; however, at the cellular level, starvation is usually characterized by a decrease in cellular metabolism [4,8]. This is a common and severe stress for animal survival [9], causing metabolic stress due to metabolic changes for higher energy production [10]. Starvation dramatically changes the liver transcriptome, upregulating genes involved in gluconeogenesis and in uptake, oxidation, storage and mobilization of fatty acids, and downregulating genes involved in fatty acid synthesis, fatty acid elongation/desaturation and cholesterol synthesis [11]. Many genes have been identified as clustered in each metabolic pathway (fat metabolism, protein, glycogen and glycogenogenesis, oxidative phosphorylation) [8,12,13,14,15,16,17,18,19,20,21]. The cellular processes are controlled by a set of interacting molecules whose activity and levels are often co-regulated or co-expressed [22].

In system genetics, the hubs are referred to as genes with the highest connectivity identified within modules and reconstructed by different network-based methods, e.g., COEX and PPI. Hub-causative genes (CST) are upstream genes affecting others and must be identified by probabilistic methods, e.g., BN. Thus, for the first time, we used the hub-CST terms to name shared upstream genes (CST genes) with the highest connectivity (hub) between BN, COEX and PPI. In PPI, only CST genes affecting fish starvation are introduced. In PPI and COEX, relationships are represented as inter- and intra-module connections [23], while in BN, cause–effect relationships can also be learned [24]. The aim of aquacultural transcriptomics research is generally to elucidate the impacts of feeding [25]. DNA microarrays allow researchers to closely follow the metabolic changes caused by starvation [18]. A number of transcriptomics starvation studies on different tissues of fish species are presented in Table 1. To date, a few DNA microarray data on the liver of starved zebrafish have been submitted to the GEO database (30 March 2022) [8,12]. Intestine, gut and muscle tissues have been investigated by RT–qPCR, DNA microarray or RNA-seq in previous studies [7,26,27,28,29]. This study aimed to use an integrated bioinformatics approach to identify a common list of modules and hub-CST genes in the zebrafish transcriptome. We envision that the present study could be the basis of future studies including functional gene analysis and regulatory mechanisms in zebrafish management systems.

## 2. Materials and Methods

### 2.1. Data and Methods

First of all, we searched datasets focusing on the gene expression profile related to starvation and feed deprivation treatment, including “starvation, feed deprivation, fasting, delayed feeding, fish, Zebrafish and liver” and a combination of these words, via the GEO database (https://www.ncbi.nlm.nih.gov/geo/, accessed on 1 June 2021). According to these selection criteria, the gene expression profiles of GSE11107 and GSE112272 datasets were found. More detailed information about the data used in this study is presented in Table 2. GEOquery R software (version “4.2”) was used to download gene expression data [38].

The overall workflow used in this study is presented in Figure 1. The probe IDs were converted to official gene symbols using corresponding data from each dataset. The maximum value of the probes was used as the expression level for the subsequent analysis. The probes that did not map to any official gene symbols were removed from the final expression matrix. In cases where several probes were mapped to the same gene names, the interquartile range (IQR) of the gene expression values in the MetaDE R package (version “4.2”) was considered as the final value of that gene [39]. Identification of DEGs was based on mean intensities of 0.2 and expression variance of 0.4. The Random Effect Model (REM) method of MetaDE with FDR < 0.01 was used to integrate related samples from GSE11107 and GSE112272 datasets [39].

### 2.2. Module Detection

The CEMiTool R package (version “4.2”) was used on the final meta-analyzed matrix to find COEX modules and hubs in the livers of starved fish. Based upon initial results, the best soft threshold (beta value of 16) for COEX module detection was chosen. Interaction data used for COEX reconstruction pertaining to DEGs were downloaded from the REACTOME database [40]. COEX hubs were identified based upon adjacency parameters, unsigned network-type and signed topological overlap matrix [22]. For COEX hub and module detection, the *p*-value was filtered at *p*-value ≤ 0.5, and an accepted R-squared interval between subsequent beta values (eps) was considered to be 0.1. For PPI module detection, the STRING database (http://string-db.org/, accessed on 1 September 2021) was used to identify PPI interactions between the proteins encoded by DEGs in starvation. Moreover, to detect hub clustering modules in the PPI network, module analysis utilizing the Molecular Complex Detection (MCODE) app with default parameters (degree cutoff = 2, cluster finding = haircut, node score cutoff = 0.2, k core = 2 and max depth = 100) in Cytoscape was performed [41]. The Venn diagram presenting the final hub-CST genes affecting starvation in the liver of zebrafish was extracted with InteractiVenn (http://www.interactivenn.net/, accessed on 12 October 2021) [42].

### 2.3. CST Gene Detection

In order to reconstruct the probabilistic network and detect causal relationships between the DEGs within the PPI and COEX modules, the bnlearn R package (version “4.2”) was used [24]. BIC score was considered as the default score for network structure reconstruction. With 50 bootstrap replicates, the model averaging method of the hill climbing algorithm was used to reconstruct the final probabilistic network. In order to ensure the relationships between the extracted genes, the threshold for acceptance and the power of a relationship between two genes, edge strength was considered to be 50%. Reconstructions of BNs were performed in two steps: BNs for genes within PPI modules and BNs for genes within COEX modules. Using the cytohubba extension of Cytoscape and degree topological parameters, CST genes were identified [43]. Graphical representation of BN within PPI and COEX modules was performed with Cytoscape [44]. For downstream analysis, the DAVID database (https://david.ncifcrf.gov/, accessed on 18 October 2021) was used to comprehensively analyze KEGG pathways of these DEGs and identify hub-CSTs [45]. Hubs were classified based on biological processes by Panther online software (http://pantherdb.org/geneListAnalysis.do/, accessed on 25 October 2021) [46]. Both *p* < 0.05 and FDR < 0.05 were considered statistically significant.

## 3. Results

The final expression matrices of meta-analyzed GSE11107 and GSE112272 datasets consist of 17 rows (samples) and 379 genes, which were successfully downloaded and are presented in Appendix A. Due to the small number of samples in this study, as well as meta-analysis of two datasets, we used the bootstrap mode of the hill climbing algorithm to reconstruct the BN structure. Hill climbing, a heuristic search algorithm, is suited for complex mathematical optimization problems. Measuring the degree of confidence in a particular a Bayesian network is a key problem in the inference of network structure. These results are from two different datasets that potentially encompass different fish genotypes and different treatments. Any particular effects due to these factors would need to be investigated in a larger, much more comprehensive study when more data become available. However, our meta-analysis will provide information on the overall identification of hub genes associated with starvation.

### 3.1. PPI Network Reconstruction

The identified genes within the PPI modules are presented in Table 3. Fourteen different modules were found, with Modules 1 and 2 having 20 and 32 genes, respectively. Modules 3 to 5 had four genes, Modules 6 to 13 had three genes and Module 14 had six genes. In Module 1, NDUF, ATP, COX, UQCR and SDH genes were identified, and, in Module 2 PSM, RPL, EIF and AGC genes were highlighted. Thus, the modules with the highest number of genes (Modules 1 and 2 with 20 and 32 genes, respectively) were selected for further analysis.

Figure 2 shows an interactive PPI network derived from DEGs associated with starvation. As can be inferred from Figure 2, a modular structure was found, indicating that genes within modules have similar functions. This was confirmed by GSEA and KEGG pathway analysis in subsequent analyses.

Figure 3 shows the inter-modular genes for DEGs for Module 1 (green) and Module 2 (pink). The 20 genes within Module 1 were ndufb10, ndufa9, uqcrb, uqcrq, ndufa6, cox7c, ndufa10, atp5h, uqcrc1, sdhc, ndufs4, ndufs1, ndufab1a, ndufs7, ndufs5, atp5f1, ndufb8, ndufb5, cox5aa and ndufa5. The most significant genes within Module 1 were nduf, cox and uqcr, contributing to oxidative phosphorylation. The 32 genes in Module 2 were psmd6, psma4, eif3d, rps3, shfm1, atp5a1, psmb3, mdh2, psmb5, psmb7, eef1g, pomp, psmd7, rpl27, psmb1, uchl5, psmc5, eif3m, rpl3, eif3i, rps27.1, rpl5a, eif2s2, psmb4, rpl22, rpl10, rpsa, psmc1a, mrpl24, rps26, psmc1b, zgc:136826 and psma6a. Intramodular genes of Module 2 were from the eif, psmb, rpl and rps families contributing to the transcriptional process, and ribosomal structures controlling metabolism and catabolism of amino acids during 21 days of starvation. However, it is not clear which genes are most significant and influence the others and the CST relationship between genes of PPI modules could not be determined; thus, we utilized a BN method to discover the intramodular casual relations within PPI modules.

The reconstructed BN from genes within modules of the PPI network from Module 1 (right) and Module 2 (left) are shown in Figure 4. The casual relationship was inferred from the direction of connections. For Module 2, a denser network structure was seen, represented by more genes. For finding CSTs within a module and considering the different number of connections to and from a gene, we recommend finding CSTs based on degree topological parameters.

In Figure 5, the identified CST genes (red and orange circles) in 2 PPI network modules are shown. In the first module, *rpl* and *rps26*, and in the second module *psmb4*, *ndufa6* and *ndufs5* were identified as CST genes affected by starvation in zebrafish. As pointed out earlier, these genes contribute to oxidative phosphorylation and transcriptional and ribosomal structures. Thus, it can be inferred that starvation may trigger hormonal and neuronal pathways leading to protein breakdown in the liver to meet body demands.

### 3.2. COEX Network Reconstruction

In Figure 6, the mean-variance trend (a), beta-r2 detection (b), sample tree (c) and histogram (d) of DEGs are shown.

In Figure 6a, the red trend line of mean expression versus variance is depicted, indicating a reasonable trend for our identified DEGs. In Figure 6b, the best soft thresholds (16) for gene co-expression network reconstruction is identified. The beta value is a parameter that lies in the core of the weighted gene COEX network analysis [22]; in our data, 16 was appropriate for COEX network reconstruction. The higher the β value, the lower its mean connectivity, with lower β values being of more interest than higher values, as long as their R2 values are similar [22]. In Figure 6c, the sample tree of our meta-analyzed data (control and starved) is presented according to their class. In Figure 6d, a histogram of meta-analyzed expression data is presented. The expression profiles of DEGs are presented in Figure 7. Samples relating to each group are distinct and shown in different colors (red: control; green: starved).

The final reconstructed COEX matrices are presented in Figure 8. Two different modules were found. Based on adjacency parameters, three different kinds of hubs (blue, green and red) were found in each module. A list of genes within COEX Module 1 (344 genes) and Module 2 (33 genes) are presented in Appendix A. Due to finding leader genes in the COEX modules, we also used a similar method (BN reconstruction) to the one we used on the PPI modules.

Figure 9 shows the CST relationship reconstructed by BN algorithms for two identified modules (Module 1: left; Module 2: right) of the COEX network. Only connected genes are shown in Figure 9. Denser structure was predicted for Module 1 due to the higher number of intramodular genes. The bootstrap, model averaging method, threshold for arc direction and strength will affect the number of genes and their connections in the BN.

The identified CST genes from the reconstructed BN form modules obtained from the COEX network and are illustrated in Figure 10. The top CST genes in the two COEX modules were from the *zgc* gene family, which is different from the PPI network module results. Thus, the results of the two networks are not the same. To better understand the molecular and metabolic mechanisms underlying the PPI and COEX modules, we performed GSEA and KEGG pathway analyses on the learned PPI and COEX modules, individually.

In Table 4, a list of hub and CST genes identified by different reconstruction network methods (BN on DEGs, PPI and COEX network) are presented.

It can be seen from Table 3 that some genes can be inferred to be either hub or causative. In Module 1, skp1, atp5h and ndufs5, and in COEX Module 2, ndufb10, rpl5b, zgc:193613, zgc:123327, zgc:123178, wu:fc58f10, zgc:111986, wu:fc37b12, taldo1, wu:fb62f08, zgc:64133 and acp5a were identified as hub-CST genes in the liver of zebrafish affected by starvation. A Venn diagram of shared genes between the PPI, BN and COEX networks is presented in Figure 11.

### 3.3. KEGG Pathway

In Table 5, KEGG pathways associated with DEGs from Module 1 and Module 2 from the PPI network are shown.

Genes within Module 1 of the PPI networks play a role in oxidative phosphorylation, metabolic pathways and cardiac muscle contraction pathways, while genes within Module 2 were associated with the proteasome and ribosome. The KEGG pathways of DEGs in Module 1 from the COEX network are presented in Table 6.

### 3.4. GSEA

Biological processes of identified PPI network modules in the liver of zebrafish affected by starvation are shown in Figure 12. In PPI Module 1, genes were active in localization, metabolic, cellular process and response to stimulus processes, while genes within PPI Module 2 involved cellular and metabolic processes, but also biological regulation processes.

Biological processes associated with genes from Module 1 (upper graph) and Module 2 (lower graph) from the COEX network are shown in Figure 13.

Genes within COEX Module 1 mainly contribute to metabolic processes, cellular processes, localization and biological regulation, while in Module 2, genes are mainly involved in binding, catalytic, structural molecule and translation regulator activities.

## 4. Discussion

With the advent of new genomic tools, it is expected that new genes and processes controlling key traits of starvation will be recognized and characterized. We used meta-analysis for integration of two DNA microarray datasets—GSE1107 and GSE112272. Meta-analysis will increase the experimental sample size and maximize the statistical power, reducing the probability of false-negative results [47,48]. Therefore, two microarray datasets obtained from GEO relating to starvation in liver tissues of zebrafish were selected to identify reliable hub-CST genes. Considering the different platforms used (Affymetrix for GSE11107; Agilent for GS112272), soft files (preprocessed data) of these datasets were used for analysis. Modification of biochemical profiles included transition from the anabolic to catabolic states. In the short term, starvation reduces glycogen and fat deposits. The zebrafish (*Danio rerio*) has high-resolution genetic maps available, a large number of offspring, ease of breeding, short life span, small size, low husbandry costs, whole genomic sequence availability, a transparent embryo, and is easily mutagenized using; therefore, it has been widely used in developmental, immunological, drug discovery, physiological, toxicological, nutrigenomic, and recently cancer and starvation studies [4,5,49,50]. *Nduf, atp, cox, uqcr* and *sdh* are gene families found in Module 1 of our PPI network In Module 2 of the PPI network, *psm, rpl, eif* and *agc* were the most significant gene families. The role of each of these genes in the study performed by Drew et al. (2008) has been discussed in more detail [12]. 

After reconstructing the BN, two genes were identified—*ndufas5*, and *ndufa6*—within Module 1, with *rps26, rpl10* and *psmb4* having influence on other genes in these PPI modules. The biological process analysis suggested that the upregulated genes were implicated in multiple processes, with the most important pathways of the PPI modules being oxidative phosphorylation, metabolic and cellular processes, ribosome and proteasome. Rpl and rps gene families are genes encoding large and small ribosomal subunits contributing to protein synthesis and degradation, the inflammatory immune system and developmental growth in the embryo [48]. Ribosomal structure is conserved in all life forms, and Rpl and rps are known as standard housekeeping reference genes for multi-organism comparisons [48,51]. The first tangible response of the liver to starvation is body mass [8], indicating rpl and rps families are activated for protein degradation after 21 days of starvation in zebrafish. Furthermore, KEGG results identified ribosomal and proteasome pathways following starvation. Studies have shown that expression of rpl and rps gene families is affected by amino acid starvation [52]. Metabolic response to starvation includes the induction of proteolytic enzyme activities and energy consumption [3]. Starvation causes protein catabolism and is accompanied by the depletion of carbon and energy sources from liver and the stimulation of gluconeogenesis from amino acids [3]. Protein degradation by autophagy is an important adaptive mechanism in fish that allows them to survive nutritional starvation [3]. However, autophagy remains constant, increasing slightly or decreasing during starvation depending on the circumstances [3]. Expression patterns of several proteins related to fatty acid and amino acid metabolism also suggested the utilization of non-carbohydrate resources for energy during starving conditions [5]. Proteins with chaperoning and antioxidative roles such as glucose-regulated protein, paraxonase and heat-shock protein were also upregulated in starvation conditions [5]. The Nduf family (NADH dehydrogenase) includes genes involved in the respiratory chain of the inner membrane of mitochondria and contributes to lipid metabolism, binding and transport [12]. During starvation, in the absence of regular feeding and unavailability of glucose and glycolysis, other metabolic cycles such as fatty acid beta-oxidation are activated; thus, oxidative phosphorylation, fatty acid metabolism and degradation are among meaningful KEGG pathways and biological processes. 

The sample size of our study was quite small, so the examination of further DNA microarrays and NGS data will be essential in supporting findings from this study, in order to investigate the different regulatory mechanisms in zebrafish subjected to starvation. Our results indicates the body’s need to produce energy in the absence of glucose (glycolysis) or to produce glucose (gluconeogenesis, beta oxidation or pentose phosphate) for tissue consumption. Previous research showed that starvation causes oxidative stress and mitochondrial malfunction in fish as a result of mitochondrial breakdown energy metabolism mRNA under expression. [28]. Within gene modules identified by COEX, the most meaningful pathways were fatty acid metabolism and degradation, suggesting that *elovl5, acat2, acadm, acox1, cpt2, fads2, hadh, scd acat2, acadm, acox1, aldh9a1b, cpt2* and *hadh* have a role in loss of body mass in zebrafish after 21 days of starvation. This finding is in accordance with some previous studies [36]. The PPAR signaling pathway is highly expressed in adipose tissue and regulates adipogenic and lipogenic pathways [53]. Other studies have shown that mRNA expression of the peroxisome proliferator-activated receptor (PPAR) was upregulated by short-term starvation through activation of lipolysis-related genes, lipid uptake-related genes and PPAR [30]. Some genes during starvation contribute to antibiotic biosynthesis (*dao.1, acat2, acadm, aldh9a1b, fntb, gapdh, hadh, idh1, idh2, pklr, zmpste24*) pathways which is in agreement with previous findings [54] demonstrating that short-term starvation prior to infection could be beneficial in obtaining better capability to battle against some infections in red sea bream [54]. Cortisol, aspartate aminotransferase, alanine aminotransferase and leptin play a major role in fish metabolism during starvation [7,10]. Here, we used a set of comprehensive bioinformatics tools to detect the genes potentially involved in starvation by comparing different groups of zebrafish. Furthermore, the molecular mechanisms associated with the starvation-related genes affected by feed deprivation of zebrafish should be further confirmed by functional validation experiments.

## 5. Conclusions

In this study, starvation was investigated by integrated bioinformatics and system genetics methods. Gene ontology and enrichment of meta-analyzed DEGs were studied individually. For this purpose, PPI and COEX networks were reconstructed; then, a BN was reconstructed within the aforementioned modules to find hub-CST genes affected by starvation in zebrafish. The results of this study also showed that BNs are useful to determine the direction of relationships between genes (CST). The hub-CST genes identified in this study are a good basis for studying starvation in zebrafish. If the hub-CST genes affected by starvation are also confirmed in vitro and in silico, their allelic frequencies could be manipulated using other Hardy–Weinberg equilibrium disturbance methods (migration, mutation and selection) and CRISPR-Cas9 technology.

## Figures and Tables

**Figure 1 animals-12-02724-f001:**
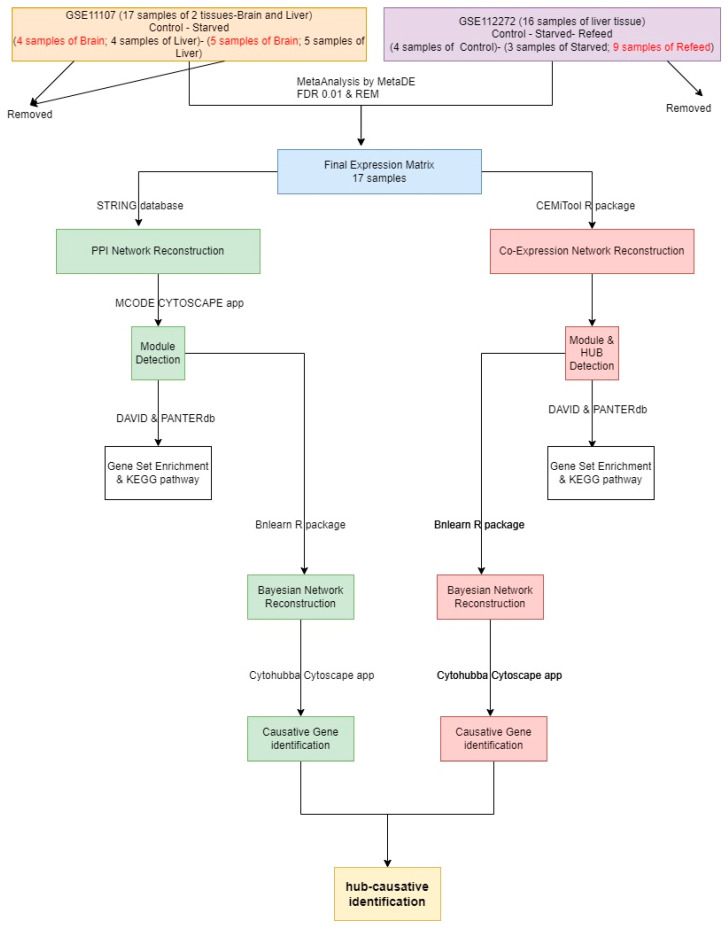
Pipeline used for investigating starvation on zebrafish in this study.

**Figure 2 animals-12-02724-f002:**
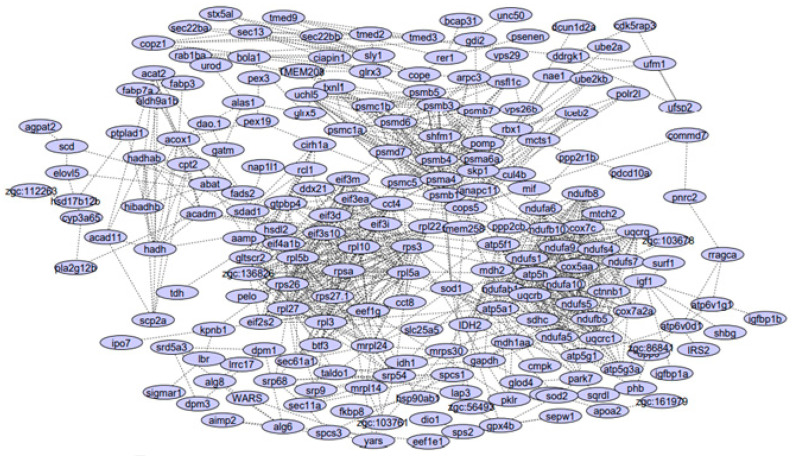
PPI network derived from DEGs affecting starvation in zebrafish.

**Figure 3 animals-12-02724-f003:**
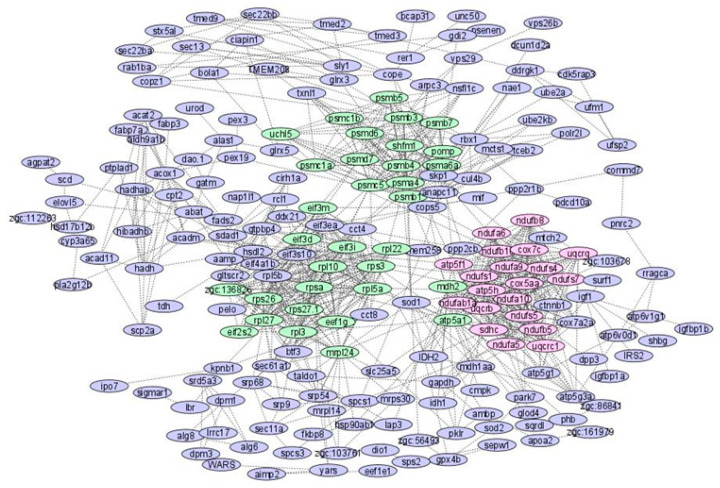
Two major modules of the PPI network (Module 1: green; Module 2: pink).

**Figure 4 animals-12-02724-f004:**
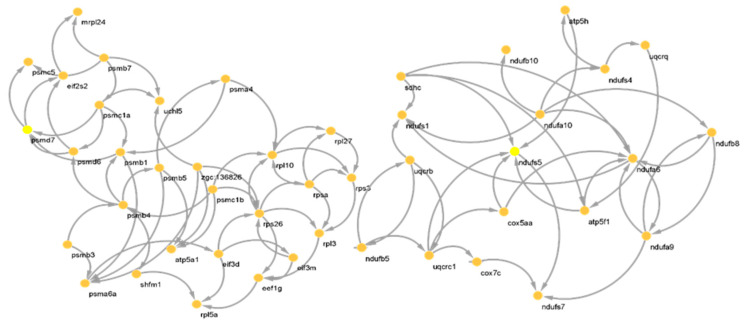
Reconstructed Bayesian network (causative relationship among genes) of genes within Module 1 (**right**) and Module 2 (**left**) of the PPI network.

**Figure 5 animals-12-02724-f005:**
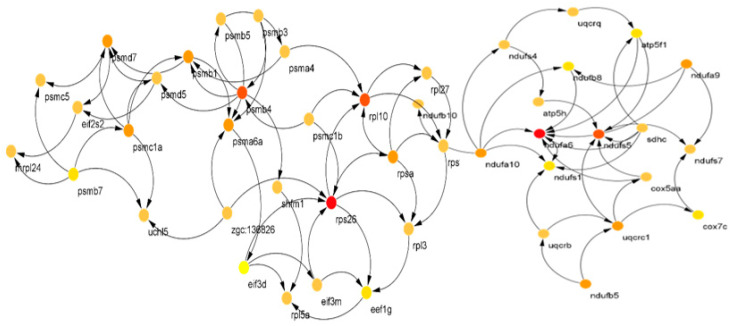
Identified causative genes (red, yellow and orange circles) from reconstructed Bayesian network of genes within Module 1 (**right**) and Module 2 (**left**) of the PPI network in Figure 4. Only connected genes are shown.

**Figure 6 animals-12-02724-f006:**
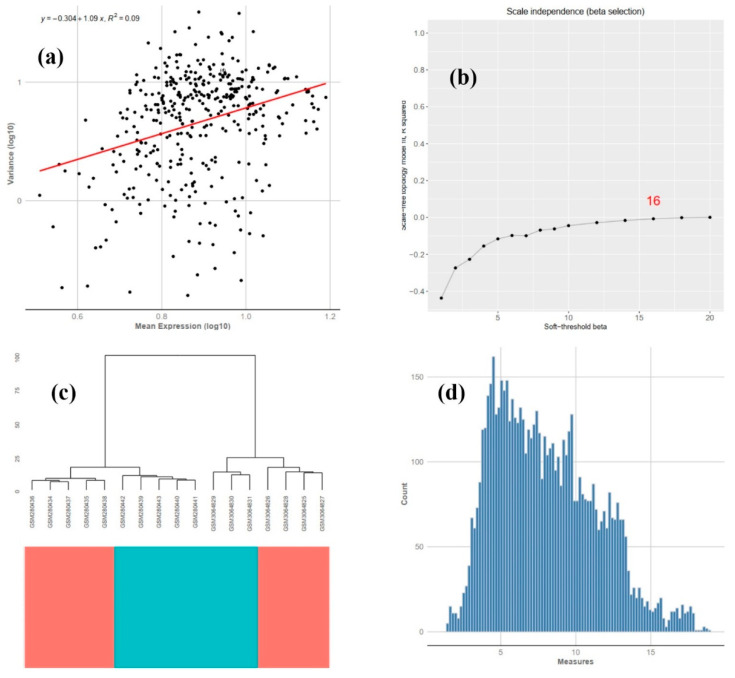
Mean-variance trend (**a**), beta-r2 detection (**b**), sample tree (**c**), histogram (**d**) of DEGs affected by starvation in zebrafish.

**Figure 7 animals-12-02724-f007:**
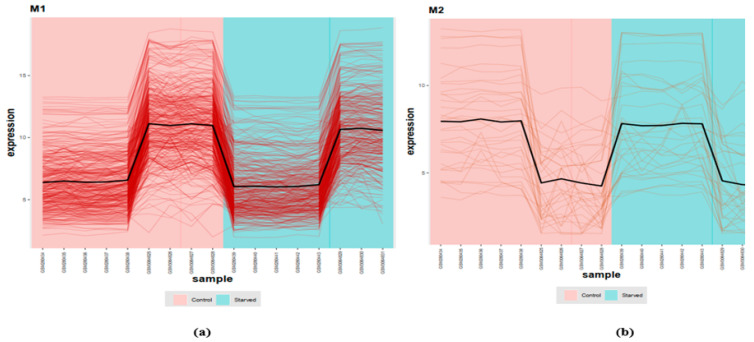
Expression profile of intra-gene modules (Module 1 (M1): **a**; Module 2 (M2): **b**) affected by starvation in zebrafish (red: control; green: starved).

**Figure 8 animals-12-02724-f008:**
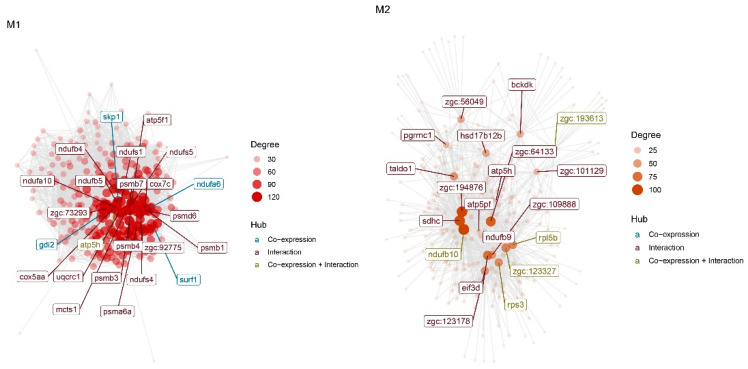
Reconstructed co-expression networks on final meta-analyzed matrix and identified Module 1 (**left**) and Module 2 (**right**).

**Figure 9 animals-12-02724-f009:**
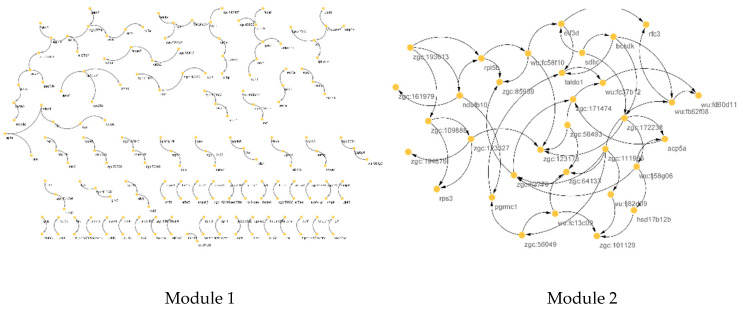
Reconstructed Bayesian network for identified modules (Module 1: **left**; Module 2: **right**) of co-expression network. Only connected genes are shown.

**Figure 10 animals-12-02724-f010:**
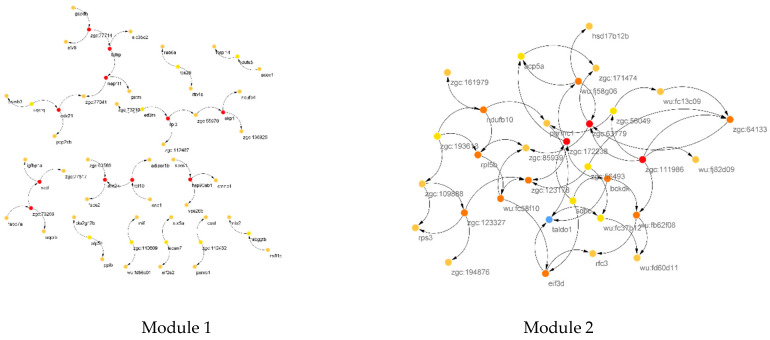
Reconstructed co-expression networks on final meta-analyzed matrix and identified Module 1 (**Left**) and Module 2 (**Right**).

**Figure 11 animals-12-02724-f011:**
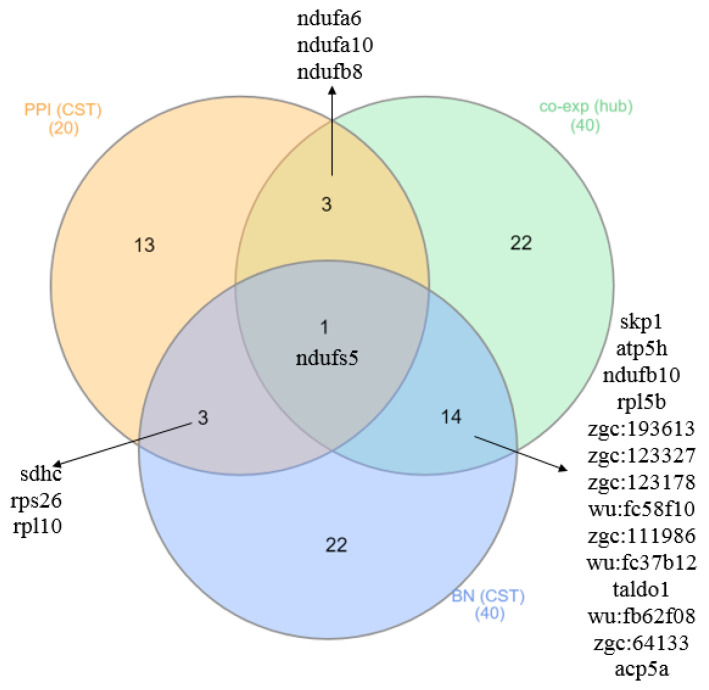
Shared and hub-causative genes between PPI, BN and COEX networks in the liver of zebrafish affected by starvation.

**Figure 12 animals-12-02724-f012:**
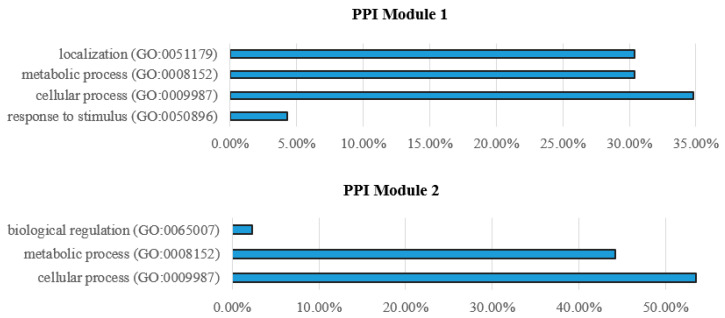
Biological processes of identified PPI network Module 1 (**upper graph**) and Module 2 (**lower graph**) in the liver of zebrafish affected by starvation.

**Figure 13 animals-12-02724-f013:**
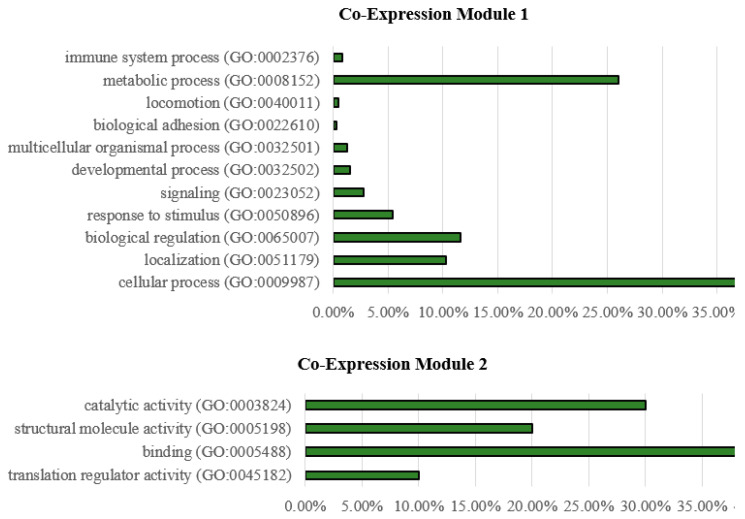
Biological processes of genes from Module 1 (**upper graph**) and Module 2 (**lower graph**) from the co-expression network in the liver of zebrafish affected by starvation.

**Table 1 animals-12-02724-t001:** Summary of transcriptomics studies performed to investigate starvation in fish.

Reference	Fish	Tissue	Duration	Platform	Endpoints
[18]	Rain bow trout	Liver	21 d	DNA microarray+ RT-PCR	Upregulation of 20S proteasome and calpain
[12] (GSE11107)	Zebrafish	Liver+ Brain	21 d	DNA microarray + qRT-PCR	Downregulation of metabolic activity, lipid metabolism, protein biosynthesis, proteolysis, cellular respiration and increased gluconeogenesis genes
[8] (GSE112272)	Zebrafish	Liver	21 d + 15 d refeeding	DNA microarray	Upregulation of TCA cycle and oxidative phosphorylation processes
[26] (GSE87704)	Zebrafish	Intestine	21 d	RNA-seq	Upregulation of ribosome biogenesis; downregulation of antiviral immunity and lipid transport genes
[27]	Zebrafish	Gastrointestinal	1, 2 & 5 d	DNA microarray + qRT-PCR	Downregulation of CCK, GRP and GHR
[30]	Zebrafish	Kidney	21 d	qRT-PCR	Upregulation of oxidative stress, catalase and superoxide dismutase genes
[29]	*S. hasta* specimens	Intestine	3, 7 & 14 d	qRT-PCR	Downregulation of stearoyl-CoA desaturase 1 diminishing lipid biosynthesis, as well as upregulation of lipolysis and fatty acid transport.
[31]	Gilthead sea bream (*Sparus aurata*)	Liver and skeletal muscle	23 d	RNA-seq	Upregulation of OXPHOS, cytochrome c oxidase families and SLC25A6
[28]	Chinese perch *Siniperca chuatsi*	Intestine	0, 7, and 14 d	RT–qPCR	Upregulation of ROS and MDA
[7]	Nile tilapia, *Oreochromis niloticus*	Intestine	14 and 21 d	RT–qPCR	Upregulation of antioxidant gene expression; downregulation of leptin
[4]	Zebrafish	Liver	70 d	RT-PCR	Downregulation of genes involved in fatty acid metabolism (elovl5, fads2, cpt1-β, acox1, acadvl, fabp1a and fabp7a)
[25]	Masu salmon	Liver and gut	3 d	RT–qPCR	DEG involved in fatty acid and carbohydrate metabolism
[6]	Brown trout	Blood	42 d	RT–PCR	Downregulation of genes involved in the elongation, desaturation and fatty acid oxidation pathways (except Δ6fadc); upregulation of pparα, pparγ and pparß
[32]	Chinese perch	Muscle	2 & 5 d	RT–qPCR	Upregulation of antioxidant-related signaling genes, Nrf2 and S6K; downregulation of Keap1
[33]	Rainbow trout	Muscle	21 d	RT–qPCR	Upregulation of genes in the ubiquitin-proteasome, lysosomal, and calpain- and caspase-dependent pathways
[34]	Atlantic Salmon	Gastrointestinal tract	4 d	RT-qPCR	Downregulation slc15a1a and slc15a1b and with significantly lowered slc15a1a
[35]	Mozambique tilapia	Intestine	14 d	RT-qPCR	Downregulation of slc6a19a expression
[36]	Zebrafish	Larvae	3 d	RNA-seq	DEG of growth regulation (i.e., DNA replication and cell cycle), energy metabolism (i.e., glycolysis/gluconeogenesis and fatty acid metabolism) and antioxidant defenses
[37]	Zebrafish	Intestine and gut	2 & 5 d	Western blot	Downregulation of PepT1 and CCK8

**Table 2 animals-12-02724-t002:** Details for datasets used for meta-analysis.

Datasets	Samples	Tissue	Species	Platform	Number ***	Reference
GSE11107 *	GSM280434-GSM280443	Liver	Zebrafish	GPL1319 affymetrix	10 (5/5)	[12]
GSE112272 **	GSM3064825-GSM3064831	Liver	Zebrafish	GPL14664 Agilent	16(4/3)	[8]

* The total number of samples was 18, eight of which were related to brain tissue, which was not examined in this study. ** The total number of samples was 16, three of were the control and four of which were the starved group considered for meta-analysis. *** Number of samples (control/starvation).

**Table 3 animals-12-02724-t003:** List of genes in identified PPI modules affected by starvation in zebrafish.

Module 1	*ndufb10, ndufa9, uqcrb, uqcrq, ndufa6, cox7c, ndufa10, atp5h, uqcrc1, sdhc, ndufs4, ndufs1, ndufab1a, ndufs7, ndufs5, atp5f1, ndufb8, ndufb5, cox5aa, ndufa5*
Module 2	*psmd6, psma4, eif3d, rps3, shfm1, atp5a1, psmb3, mdh2, psmb5, psmb7, eef1g, pomp, psmd7, rpl27, psmb1, uchl5, psmc5, eif3m, rpl3, eif3i, rps27.1, rpl5a, eif2s2, psmb4, rpl22, rpl10, rpsa, psmc1a, mrpl24, rps26, psmc1b, zgc:136826, psma6a*
Module 3	*spcs1, srp9, srp68, sec11a*
Module 4	*tecb2, anapc11, cops5, cul4b*
Module 5	*ufsp2, cdk5rap3, ddrgk1, ufm1*
Module 6	*eif4a1b, eif3s10, eif3ea*
Module 7	*sdad1, gtpbp4, ddx21*
Module 8	*cirh1a, gltscr2, rcl1*
Module 9	*abat, aldh9a1b, hibadhb*
Module 10	*stx5al, sec22bb, sec22ba*
Module 11	*gpx4b, zgc:56493, sod2*
Module 12	*zgc:103761, mrpl14, mrps30*
Module 13	*mtx2, dnajc11, chchd3*
Module 14	*gatm, bola1, glrx5, dao.1, ciapin1, alas1*

**Table 4 animals-12-02724-t004:** List of hub and CST genes identified by different network reconstruction methods by meta-analysis of GSE11107 and 112272 liver datasets affected by starvation of zebrafish.

Method	Hub/CST genes
	**PPI network**
CST of M1	*ndufa6,* *ndufs5, uqcrc1,* *ndufa10, atp5f1, ndufs1, sdhc, uqcrb,* *ndufb8, ndufs7*
CST of M2	*rps26,* *rpl10, psmb4, psmc1a, psmb1, psmd7, psma6a, rpsa, psmb7, eef1g*
	**Coexpression network**
**By CEMiTool (adjacency)**	
Hub of M1	*skp1,* *atp5h, gdi2, surf1, ndufa6,* *ndufs5, ndufb5, cox5aa, alg8, nnt, rab11b,* *ndufa10, zgc:73210, glod4, hadh, pdcd10a,* *ndufb8, slc30a5, rhoaa, psmc1b*
Hub of M2	*ndufb10,* *rpl5b,* *zgc:193613,* *zgc:123327, rps3, zgc:109888, zgc:161979, pgrmc1,* *zgc:123178, zgc:194876, zgc:85939,* *wu:fc58f10,* *zgc:111986, wu:fd60d11,* *wu:fc37b12,* *taldo1, wu:fj82d09,* *wu:fb62f08,* *zgc:64133,* * acp5a *
**By BNLEARN (degree)**	
CST of M1	*skp1, hsp90ab1, zgc:73269, scd,* *rpl10, tollip, zgc:77714, ddx21, rpl3, nap1l1, ube2k, zgc:110609, rabggtb, uqcrq, eif3m, tspan7,* *atp5h, zgc:112432* *, rps26,* * ndufs5 *
CST of M2	*zgc:172238,* *zgc:111986, zgc:63779,* *zgc:123327,* *taldo1, eif3d, bckdk,* *zgc:64133,* *wu:fb62f08,* *zgc:123178, wu:fj58g06,* *ndufb10,* *rpl5b,* *wu:fc58f10, zgc:56493,* *wu:fc37b12, sdhc, zgc:56049,* *acp5a,* * zgc:193613 *

Red fonts indicate hub and causative genes in Module 1. Green fonts indicate hub and causative genes in Module 2. Purple fonts indicate hub and causative genes in Module 2 of PPI and Module 1 of BNLEARN networks.

**Table 5 animals-12-02724-t005:** KEGG pathways of DEGs from Module 1 and Module 2 from the PPI network in the liver of zebrafish affected by starvation.

**Module 1**			
**Term**	** *p* ** **-Value**	**Benjamini**	**Genes**
Oxidative phosphorylation	1.70 × 10^26^	1.00 × 10^25^	*atp5f1, atp5h, ndufa10, ndufa6, ndufb5, ndufb10, ndufb8, ndufs1, ndufs4, ndufs5, ndufs7, coxaa, cox7c, sdhc, uqcrb, uqcrc1, uqcrq*
Metabolic pathways	6.70× 10^11^	2.00 × 10^10^	*atp5f1, atp5h, ndufa10, ndufa6, ndufb5, ndufb10, ndufb8, ndufs1, ndufs4, ndufs5, ndufs7, coxaa, cox7c, sdhc, uqcrb, uqcrc1, uqcrq*
Cardiac muscle contraction	1.40 × 10^4^	2.90 × 10^4^	*cox5aa, cox7c, uqcrb, uqcrc1, uqcrq*
**Module 2**			
**Term**	** *p* ** **-Value**	**Benjamini**	**Genes**
Proteasome	9.10 × 10^20^	1.10 × 10^18^	*psmc1a, psmc1b, psmc5, psmd6, psmd7, pomp, psma4, psma6a, psmb1, psmb3, psmb4, psmb5, psmb7, shfm1*
Ribosome	1.30 × 10^8^	7.50 × 10^8^	*mrpl24, rpl10, rpl22, rpl27, rpl3, rpl5a, rps26, rps27.1, rps3, rpsa*

**Table 6 animals-12-02724-t006:** KEGG pathways of DEGs from Module 1 and Module 2 from the COEX network in the liver of zebrafish affected by starvation.

Module 1			
Term	*p*-Value	Benjamini	Genes
Proteasome	6.90 × 10^9^	3.80 × 10^7^	*psmc1a, psmc1b, psmc5, psmd6, psmd7, pomp, psma4, psma6a, psmb1, psmb3, psmb4, psmb5, psmb7, shfm1*
Oxidative phosphorylation	8.40 × 10^9^	3.80 × 10^7^	*atp5f1, atp5h, ndufa10, ndufa6, ndufb5, ndufb10, ndufb8, ndufs1, ndufs4, ndufs5, ndufs7, coxaa, cox7c, sdhc, uqcrb, uqcrc1, uqcrq*
Protein export	1.70 × 10^4^	5.20 × 10^3^	*sec11a, spcs1, spcs3, srp54, srp68, srp9,*
Fatty acid metabolism	5.60 × 10^4^	1.30 × 10^2^	
Fatty acid degradation	4.20 × 10^3^	7.50 × 10^2^	*acat2, acadm, acox1, aldh9a1b, cpt2, hadh*
Metabolic pathways	5.10 × 10^3^	7.70 × 10^2^	*agpat2, hibadhb, abat, alg8, atp5a1, atp5f1, atp5h, atp6v0d1, atp6v1g1, dao.1, ndufa10, ndufa6, ndufb5, ndufb4, ndufb8, ndufs1, ndufs4, ndufs5, ndufs7, acat2, acadm, acox1, ahcy, aldh9a1b, alas1, alg6, cmpk, cyp3a65, cox5aa, cox7c, dpm1, dpm3, gapdh, gatm, hadh, idh1, idh2, lap3, nnt, pla2g12b, pklr, sps2, uqcrb, uqcrc1, uqcrq, urod,*
PPAR signaling pathway	6.90 × 10^3^	8.10 × 10^2^	*Acadm, acox1, cpt2, fabp3, fabp7a, fads2, scd,*
Peroxisome	7.20 × 10^3^	8.10 × 10^2^	*dao.1, acox1, idh1, idh2, pex19, pex3, sod1, sod2*
Valine, leucine and isoleucine degradation	9.40 × 10^3^	9.40 × 10^2^	*hibadhb abat, acat2, acadm, aldh9a1b, hadh*
Biosynthesis of unsaturated fatty acids	2.10 × 10^2^	1.90 × 10^1^	*elovl5, acox1, fads2, scd*
Ribosome	5.10 × 10^2^	4.20 × 10^1^	*mrpl24, rpl10, rpl22, rpl27, rpl3, rpl5a, rps26, rps27.1, rps3, rpsa*
Biosynthesis of antibiotics	6.80 × 10^2^	5.10 × 10^1^	*dao.1, acat2, acadm, aldh9a1b, fntb, gapdh, hadh, idh1, idh2, pklr, zmpste24,*
alpha-linolenic acid metabolism	7.40 × 10^2^	5.10 × 10^1^	*acox1, fads2, pla2g12b*

## Data Availability

The data that support the findings of this article are available from GEO database of the NCBI with accession number GSE11107 (https://www.ncbi.nlm.nih.gov/geo/query/acc.cgi?acc=GSE11107 (accessed on 9 April 2018)) and GSE112272 (https://www.ncbi.nlm.nih.gov/geo/query/acc.cgi (accessed on 23 March 2018)).

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
