# Peer review of "An Integrated Bioinformatics Approach to Identify Network-Derived Hub Genes in Starving Zebrafish"

_animals, 2022, doi:10.3390/ani12192724_

Round 1

Reviewer 1 Report (Previous Reviewer 2)

This paper reports a study of starvation-induced changes in zebrafish liver transcriptome, using a combination of bioinformatics and systems biology methods, such as Bayesian Network construction, for the analysis of co-expression and protein–protein interaction (PPI) networks derived from two gene-expression datasets.

The analysis methodology and the results are clearly described and the discussion and the conclusions are in line with the results.

A major shortcoming of this study, which the authors identify, is the small sample size.  

Some comments/suggestions:

1 - In line 68 the authors write: “CST are upstream genes affecting others (...)”. What is the meaning of “CST”? The exact meaning should be specified when it is first mentioned.

2 - In line 131 I believe “network” should be “networks”.

3 - In line 132, the sentence “In order to ensure the relationships between the extracted genes (...)” is not clear. It appears to be missing the subject. The sentence should be rephrased to improve readability.

4 - It is not clear what is meant with the sentence from lines 338 and 339. Please contextualize.

5 - The considerations regarding small sample size (line 400) should be moved from the conclusions to the discussion section. Eventually this could be combined with the consideration from lines 146-150 to create a section discussing the limitations of the study.

6 - It is not clear why the use of the bootstrap mode of the Hill Climbing algorithm is appropriate  given the small sample size and the fact this is a meta-analysis (lines 400 and 401). Please elaborate.

Author Response

Reviewer 1:

This paper reports a study of starvation-induced changes in zebrafish liver transcriptome, using a combination of bioinformatics and systems biology methods, such as Bayesian Network construction, for the analysis of co-expression and protein–protein interaction (PPI) networks derived from two gene-expression datasets.

The analysis methodology and the results are clearly described and the discussion and the conclusions are in line with the results.

A major shortcoming of this study, which the authors identify, is the small sample size.  

Response: After a comprehensive exploration of GEO, the most important and relevant microarray studies on starvation in fish were GSE11107 and GSE112272. This only provided us with 17 samples to analyze. Although this is admittedly relatively small, it still allowed us to have 9 control samples and 8 starved samples for our analysis. We found we had sufficient statistical power with these numbers and are satisfied with our conclusions.

Point:

 In line 68 the authors write: “CST are upstream genes affecting others (...)”. What is the meaning of “CST”? The exact meaning should be specified when it is first mentioned.

Response: We corrected it in the text.

Point:

In line 131 I believe “network” should be “networks”.

Response: We corrected it in the text.

Point:

In line 132, the sentence “In order to ensure the relationships between the extracted genes (...)” is not clear. It appears to be missing the subject. The sentence should be rephrased to improve readability.

Response: We paraphrased this sentence.

Point:

It is not clear what is meant by the sentence from lines 338 and 339. Please contextualize.

Response: We deleted this sentence.

Point:

The considerations regarding small sample size (line 400) should be moved from the conclusions to the discussion section. Eventually this could be combined with the consideration from lines 146-150 to create a section discussing the limitations of the study.

Response: We corrected in the text.

Point:

It is not clear why the use of the bootstrap mode of the Hill Climbing algorithm is appropriate given the small sample size and the fact this is a meta-analysis (lines 400 and 401). Please elaborate.

Response: that is a beautiful and equally thought-through remark. It is almost worldly accepted that in these sorts of studies, due to the “curse of dimensionality,” we are to face up with a false discovery rate! Using some trick, e.g., Meta-analysis (provided having reliable data handy), we try to level down FDR issue! But still, we could use some other accepted statistical tool (bootstrap) to somehow better combat small-size data. Hill Climbing, a heuristic search algorithm, is sued for mathematical optimization problems of a complex problem. Measuring the degree of confidence in a particular Bayesian network is a key problem in the inference of the network structure. In our study, by means of bnleran we generated multiple network structures by applying nonparametric bootstrap to the data and estimating the relative frequency of the feature of interest, e.g., arc strength. 

Reviewer 2 Report (Previous Reviewer 3)

The manuscript has improved substantially. The authors have addressed my concerns.

Author Response

Reviewer 2:

The manuscript has improved substantially. The authors have addressed my concerns.

Response: We would like to thank the reviewer for letting us improve our work.

This manuscript is a resubmission of an earlier submission. The following is a list of the peer review reports and author responses from that submission.

Round 1

Reviewer 1 Report

The main aim of the manuscript “Integrated Bioinformatics of Omics Data to Locate Network Derived Hub Genes in Starving Zebrafish” was to use an integrated bioinformatics approach to identify a common list of modules and hub-CST genes in zebrafish. For this purpose, two previously reported transcriptomes from zebrafish liver tissue under starvation were integrated. In addition, the authors identified hub- causative genes within modules learned by co-expression and protein–protein interaction (PPI) networks followed by Bayesian Network (BN) in the liver transcriptome of starved Zebrafish.

In this sense, in the present study, a meta-analysis, reanalyzing two sets of expression microarray data, were performed and showed the usefulness of various analysis methodologies to identify gene networks and their possible causal relationships. This approach can be applied in other species where a large number of data sets are available.

Define the first time an acronym is mentioned in the main text.

Reviewer 2 Report

This paper describes an approach to the identification of genes associated with starvation in zebreafish in coexpression and PPI networks.

While the methodological approach seems valid but I have tow major criticisms of the paper:

1 - Sample size is very small. Two studies yielding 17 samples is a very slim basis.

2 - The paper needs to be rewritten by someone with a better working knowledge of the English language.

It is not possible to assess the results properly unless number 2 is addressed.

Some specific remarks:

* The "Simple summary" is incomprehensible.

* The definitions of "hub-causative" and "hub-CST" are confusing.
  "CST" is never defined, besides this sentence: "While CST are upstream genes affecting others and must be identified by probabilistic methods e.g. BN.", which I can't understand.   

* Please standartize the language, for example, in line 37 the authors mentions the "bnlearn R package", in line 128 the "bnlearn bundle in R" is mentioned.

* I would suggest to add the versions and or access dates of the packages and databases used.

I hope I don't sound too harsh.
It is clear that a lot of workand thought has been done here, however
I believe it is mandatory to make an extensive re-write of the paper for publication.

Reviewer 3 Report

In this manuscript, Mortazavi et al. reported a re-meta-analysis of two public datasets of Zebrafish. As stated in the summary, the authors claimed that this “integrative bioinformatics” approach enables them to “explore crucial transcriptional insights” ” in an understandable way” to study Zebrafish. The whole study is very descriptive, and no functional validations at all to support the newly identified key factors. I am just not convinced this newly proposed “integrative bioinformatics” approach is very different from previous studies. It is just the routine gene expression analysis

Other minor issues.

1.     The two datasets used were generated by two very different platforms, the authors need to address how to handle these technology differences when integrating them.

2.     The Zebrafish genotypes of these datasets need to be given, and the authors need to discuss whether the different genotypes and treatments could impact the re-analysis.

3.      Fig 2, it is not clear to me that whether the factors identified in previous studies are identified in this meta study. If known factors were identified, they should be highlighted in this figure. This figure is hard to read. Does the line length mean anything? It is very hard to spot the curtail nodes/vertices in this graph.

4.     Fig 4, 5, 9 and 10, does the length mean anything. What do the arrow mean, activating?